# The Improved Biometric Identification of Keystroke Dynamics Based on Deep Learning Approaches

**DOI:** 10.3390/s24123763

**Published:** 2024-06-09

**Authors:** Łukasz Wyciślik, Przemysław Wylężek, Alina Momot

**Affiliations:** 1Department of Applied Informatics, Faculty of Automatic Control, Electronics and Computer Sciences, Silesian University of Technology, 44-100 Gliwice, Poland; alina.momot@polsl.pl; 2Healthcare Solutions Department, NubiSoft, 44-100 Gliwice, Poland; przemyslaw.wylezek@nubisoft.pl

**Keywords:** keystroke dynamics, variable text, user identification, deep learning

## Abstract

In an era marked by escalating concerns about digital security, biometric identification methods have gained paramount importance. Despite the increasing adoption of biometric techniques, keystroke dynamics analysis remains a less explored yet promising avenue. This study highlights the untapped potential of keystroke dynamics, emphasizing its non-intrusive nature and distinctiveness. While keystroke dynamics analysis has not achieved widespread usage, ongoing research indicates its viability as a reliable biometric identifier. This research builds upon the existing foundation by proposing an innovative deep-learning methodology for keystroke dynamics-based identification. Leveraging open research datasets, our approach surpasses previously reported results, showcasing the effectiveness of deep learning in extracting intricate patterns from typing behaviors. This article contributes to the advancement of biometric identification, shedding light on the untapped potential of keystroke dynamics and demonstrating the efficacy of deep learning in enhancing the precision and reliability of identification systems.

## 1. Introduction

In the contemporary landscape of human progress, the rapid and unprecedented growth of information technology (IT) is reshaping various domains and engaging an increasing number of individuals. Whether considering ubiquitous technological services for home entertainment, systems facilitating end users in banking, healthcare, public administration, or specialized industry-specific systems, safeguarding information and access to services remains paramount. The protection of information and service access is pivotal across diverse sectors, highlighting the critical need for the development of more and more effective and advanced authentication and authorization methods.

Diverse ways of implementing and sharing systems dictate different approaches to user authentication and identification, extending beyond traditional methods based on usernames and passwords. The overarching goal is to enhance user ergonomics while simultaneously fortifying the security levels of IT systems.

A particularly effective and promising avenue for user authentication and authorization revolves around biometric methods, eliminating the necessity for users to memorize secret tokens or possess additional devices. Biometric approaches offer not only convenience but also enable periodic or continuous user authorization, especially when safeguarding access to highly sensitive data and services.

Among the prevalent biometric methods such as fingerprint or retinal scans, there exists a distinct branch focusing on behavioral characteristics [1], including mouse movements [2] or keystroke dynamics analysis [3,4]. This approach holds significant promise, especially in systems where user interaction occurs through keyboards continuously [5,6]. Authentication and authorization can seamlessly take place without requiring additional efforts from users, making it a valuable complement to traditional password-based methods. Moreover, traditional methods of verifying the matching of username and password can be additionally extended to include aspects of the dynamics related to the way these tokens are input into the systems.

Initially, the implementation of keystroke dynamics-based authentication systems relied on statistical classification approaches or machine learning methods, including Bayesian networks and support vector machines. However, in recent times, the most notable advancements have been achieved through the application of deep learning techniques which constitute a distinct field in themselves, offering a plethora of possibilities due to the diversity of neural network layers and the multitude of base architectures employed within this domain. The flexibility provided by deep learning allows for the construction, testing, and refinement of various solutions, making it a dynamic and continuously evolving area of research. However, a significant challenge in the advancement of research in the field of keystroke-based biometric identification is the lack of a widely accessible and universally accepted benchmark dataset, against which researchers could compare and evaluate the progress of their studies, thereby enhancing their credibility and reliability. Therefore, this paper includes a literature review wherein individual authors describe their most promising results. This facilitated the identification of the most suitable benchmark dataset, upon which the results of the proposed approach presented in this article is based.

It is also worth noting that verifying the user’s identity through keystroke dynamics can serve as a complementary method used in addition to traditional ones. Therefore, the authors of this study aimed to increase the detection threshold of potential intruders, even at the cost of requiring more frequent verification using classical methods.

Further presentation of the conducted research outlined in this article is structured as follows. In the “Related Work” section, significant prior research leveraging deep neural networks for user identification based on keystroke dynamics is presented. The “Materials and Methods” section comprehensively details the benchmark dataset used and research methodologies employed in the present study. The “Results” section encapsulates the findings derived from the study, while the “Conclusions” section encapsulates the insights drawn from the results, delineates the limitations of the research, and suggests potential avenues for further exploration.

## 2. Related Work

### 2.1. Background

User identification based on keystroke dynamics is a method that was first employed in research in the late 1970s [7]. The researchers experimented to verify the hypothesis that there are certain temporal patterns distinguishing users during keyboard typing. Each participant in the experiment was asked to input three phrases. Additionally, users were requested to repeat the experiment after a 4-month interval. The timing of each key press during user input was recorded. Subsequently, the time interval between the consecutive key press combinations, referred to as digraphs, was analyzed. One of the experiment’s conclusions was the observation that the temporal characteristics of digraphs remained consistent across two sessions recorded with a four-month gap for the same users, justifying further exploration of this method for user identification.

The data analysis approach at that time was based on a purely statistical method, comparing time values between occurring digraph combinations using graphs and the distances between them. It is noteworthy that, with the evolution of the method, researchers increasingly adopted the use of trigraphs and quadgraphs, combinations of three and four consecutive key presses, respectively.

Trigraphs and quadgraphs allow for the extraction of a larger number of features from the input data, and the number of features that can be extracted depending on the number of analyzed keys is given by
(1)$$N_{c}=\frac{(2N_{k})!}{2!(2N_{k}-2)!}$$
where $N_{c}$ is the number of features and $N_{k}$ is the number of keys analyzed [8]. However, it should be taken into account that, as the size of the graph increases, the computational and memory complexity of the solution also increases, and, additionally, sometimes the effectiveness of the algorithm based on digraphs turns out to be better than trigraphs or whole words, as shown by the research described in [9].

Over time, research expanded to investigate the justification for employing alternative classifiers for user identification or verification, including methods from the machine learning domain. An independent and intensive area of study focuses on exploring the feasibility of applying keystroke dynamics based on longer and arbitrary sequences entered by users on the keyboard, rather than solely on fixed text. This approach allows for the method’s application as both a component of two-factor authentication and in the continuous authentication process.

### 2.2. Deep Learning Approaches

In the case of user classification based on typing patterns for variable text, similar to fixed text, machine learning methods and their application as classifiers are commonly chosen by researchers. In 2014, a publication [10] introduced a method for user identification based on keystroke dynamics using multilayer perceptron (MLP) neural networks. The authors extracted features from any text without restricting the nature of the key sequence used by the study participants. The dataset was collected over a 5-month period, recording sequences of key presses by 53 users during their daily computer-related activities, resulting in a resource containing an average of 18,008 key press events for each user. It can be noted that the data used in the experiment were obtained in an uncontrolled manner.

The decision model constructed by the researchers was based on multilayer perceptron (MLP) neural networks. Similarly to other studies [11,12], during the preparation of input data for the model, two features were extracted, the time between pressing and releasing a key H_time_ and the time between releasing the current key and pressing the next key UD_time_ (up to down time). The decision model relied on two neural networks, each powered by a single feature extracted from the input data. Input and output data at the neural network level were normalized using the min–max algorithm. To evaluate the system’s effectiveness for each user, the set of obtained sequences was divided into training and validation sets, with 1500 digraphs included in the training set. The remaining data for a given user were placed in the validation set. Cross-validation was applied during the model evaluation using the leave-one-out approach, repeating the experiment 53 times for the same user. As a result of the experiment, average values of false acceptance rate (FAR) coefficients were obtained at the level of 0.0152%. The average value of the false positive rate (FPR) coefficient was achieved at 4.82%, while the equal error rate (EER) coefficient was 2.46%. The results obtained by the authors can be considered above average compared to previous works in the field of personal identification based on keystroke dynamics.

Despite the high interest in machine learning methods for analyzing typing patterns in variable text, statistical methods continue to be an area of continuous research for this problem. In 2019, Ayotte et al. presented a user authentication method based on keystroke dynamics for variable text, employing statistical methods in the decision model [13]. The authors utilized the Clarkson II dataset for their experiment. During the initial data processing and feature extraction from the dataset, researchers chose to use the following features: the first key code K_1_, the second key code K_2_, and only one of the available time features of the digraph, the time between pressing the first key and pressing the second key DD_time_ (down to down time). Three statistical methods were used and compared in the study:KDE (Kernel Density Function) [14];ED (Energy Distance) [15];Kolmogorov–Smirnov test [16].

Furthermore, the study presented results for classifiers using the fusion of these methods. The authors also explored the impact of the digraph vector length on classification accuracy. Experiments were conducted for digraph sequences of different lengths: 100, 200, 500, and 1000. For a feature vector consisting of only 100 digraphs, the lowest EER coefficient, at 35.1%, was achieved for the classifier based on the fusion of KDE and ED methods. For sequences of length 200, the best result was achieved for a model based on the fusion of all three methods analyzed by the authors, i.e., KDE, ED, and Kolmogorov–Smirnov test, setting the EER coefficient at 15.3%. For a feature vector of length 500, the lowest EER value, 6.3%, was achieved for the KDE method. The lowest EER value, at 3.6%, was obtained for a feature vector of length 1000 and a classifier based on the fusion of three methods, KDE, ED, and Kolmogorov–Smirnov test.

A year later, the same authors published a research paper [17], continuing the investigations from the article [13] from 2019, where satisfactory results were obtained for the Clarkson II dataset. In reference to their previous work, a significant drawback of the proposed solution was the minimum required length of a single sequence, which was 1000. In systems requiring rapid unauthorized access detection, the application of the proposed approach was not feasible. Building on the conclusions from the previous article, the authors decided to explore other statistical methods and propose their own method called ITAD (Instance-based Tail Area Density) to reduce the required minimum sequence length for the Clarkson II dataset. Among the methods examined by the authors were the following:KDE;Manhattan Distance;Scaled Manhattan Distance;Mahalanobis Distance;Transformed Mahalanobis Distance.

For comparison, the authors extracted the following features: the first key code K_1_, the second key code K_2_, and the time between pressing the first key and pressing the second key DD_time_. Additionally, compared to the previous study, the authors extracted the following features: the time between releasing the first key and pressing the second key UD_time_ (up to down), the time between pressing the first key and releasing the second key DU_time_ (down to up), and the time between releasing the first key and releasing the second keyUU_time_ (up to up time) for the Clarkson II dataset. A comparative experiment was conducted based on the EER metric for sequences of length 50. The study compared the effectiveness of each method, also examining the impact of the feature vector size on classification accuracy for each method. According to the authors, identification based on the fusion of all features allows for a reduction in the balanced error rate (EER). For the Clarkson II dataset, the ITAD method and the fusion of all features achieved the lowest EER value among the results for each approach, at 12.3%. In another experiment, the effectiveness of the proposed classifier was examined for different sequence lengths, namely 10, 20, 100, and 200. For shorter sequences of lengths 10 and 20, a high EER error of 22.1% and 17.7% was recorded, respectively. Nevertheless, compared to the previous article [13], a significant improvement in the classification method was achieved for sequences of lengths 100 and 200, setting the EER error at 9.07% and 7.8%, respectively. Additionally, the authors decided to conduct an experiment regarding the impact of sequence length for other data, namely the Buffalo dataset [18]. For sequences of lengths 10, 20, 50, 100, and 200, the EER coefficient was 19.9%, 13.6%, 8%, 5.3%, and 3%, respectively. The results obtained by the authors for both the Clarkson II and Buffalo datasets can be considered above average compared to the results of other studies in the field of user identification based on keystroke dynamics.

An area of intensive research in recent years is the analysis of the feasibility of applying recurrent and convolutional neural networks in the context of typing pattern analysis for variable text. In 2019, Lu et al. [19] presented a binary classifier based on RNN [20] and CNN [21] neural networks. The authors used the publicly available Buffalo dataset [18]. During the dataset processing and preparation of input data for the model, the authors performed feature extraction, including K_1_, K_2_, H1_time_, H2_time_, UD_time_, and DD_time_, extracting a total of six features for a single digraph. Due to the required input format for RNN-type neural networks, the authors aggregated feature vectors, obtaining sequences, each representing a two-dimensional feature vector for consecutively occurring digraphs for a given user. In the experiment, an analysis was conducted on the impact of several variables on classification accuracy, including the following:Sequence length;Feature vector size;Neural network architecture (RNN architecture and CNN + RNN architecture, where the convolutional network precedes the recurrent neural network).

To determine the optimal sequence length, the authors examined the classification accuracy for sequences of different lengths: 10, 30, 50, 70, and 100. The experiment demonstrated that sequence length significantly influences classification accuracy. According to the authors, a sequence that is too short contains insufficient information about a user. For the experiment with a sequence length of 10, unsatisfactory results were obtained, where the values of the FRR, FAR, and EER coefficients were 16.02%, 3.48%, and 9.75%, respectively. Furthermore, the authors showed that a sequence that is too long contains noise that negatively affects identification effectiveness. For a sequence length of 100, satisfactory but not the best results for the entire experiment were achieved, where the FRR, FAR, and EER coefficients were set at 2.67%, 7.57%, and 5.12%, respectively. According to the authors, based on the lowest values of the FRR, FAR, and EER coefficients, the optimal sequence length for the Buffalo dataset is 30. For the experiment with the optimal sequence length of 30, the coefficient values were 1.95% for FRR, 4.12% for FAR, and 3.04% for EER. The authors also demonstrated that the feature set used to build the model affects classification accuracy. In the experiment, the optimal sequence length determined in the previous experiment was used, which was 30. As a result of the research, for a feature set containing only a subset of all extracted features, namely K_1_, K_2_, H1_time_, and H2_time_, the lowest classification accuracy was obtained, where the FRR and FAR coefficients were set at 12.39% and 5.96%, respectively, and the EER coefficient reached 9.17%. The highest classification accuracy was achieved in the experiment in which the model was powered by the full set of extracted features: K_1_, K_2_, H1_time_, H2_time_, UD_time_, and DD_time_, where the FRR, FAR, and EER coefficients were 1.95%, 4.12%, and 3.04%, respectively. In the last experiment, the study investigated and compared the classification accuracy depending on the neural network architecture. During the experiment, the authors took into account the conclusions from previous research, conducting studies for the optimal sequence length (30) and the optimal feature set (K_1_, K_2_, H1_time_, H2_time_, UD_time_, and DD_time_). The analysis showed that, using only recurrent neural networks as classifiers, based on GRU cells, allows achieving satisfactory results, with metric values at 4.05% for FRR, 6.01% for FAR, and 5.03% for EER. However, for the CNN + RNN architecture, higher classification accuracy was obtained than in the case of the RNN network alone. According to the authors, the convolutional layer preceding GRU units allows for the extraction of higher-order features from input data, thus improving identification effectiveness. The FRR, FAR, and EER values for the classifier with CNN + RNN architecture were 1.95%, 4.12%, and 3.04%, respectively. It should also be emphasized that the authors’ use of the public Buffalo dataset allows for a reliable comparison of results with other studies in the field of user identification based on keystroke dynamics.

A year later, Lu et al., continuing their own research [19], reanalyzed the possibility of using neural networks in the CNN + RNN architecture as a binary classifier based on typing patterns, examining the impact of various factors on model effectiveness [22]. Two public datasets, namely Clarkson II and Buffalo, were used for the studies. The authors decided to examine a series of parameters, both in the feature extraction domain and in recurrent and convolutional networks. Experiments were conducted for different network architectures and their parameters. In the first experiment, the impact of the type of recurrent network was analyzed, comparing LSTM (Long Short-Term Memory) and GRU (Gated Recurrent Unit) networks, additionally conducting an analysis for a different number of cells within each of the used layers. As a result of the experiment, the model using GRU-type recurrent neural networks with 16 units in a single layer achieved the lowest EER coefficient, at 14.28% for the Clarkson II dataset and 3.61% for the Buffalo dataset. In the second experiment, the impact of using convolutional networks for extracting higher-order features was examined. The lowest balanced EER value for this experiment was obtained for the CNN + GRU (16) network variant, reducing the EER error compared to the previous experiment to 2.67%. In the next analysis, the impact of the kernel size parameter on user verification accuracy was examined. It turned out that the optimal value of this parameter for both datasets, Clarkson II (with an EER of 6.61%) and Buffalo (with an EER at the level of 2.67%), is 2. In the fourth study, the accuracy of binary classification was analyzed depending on the sequence length, obtaining an optimal sequence length of 50 for both datasets and recording an EER error value at 5.97% for the Clarkson II dataset and 2.36% for the Buffalo dataset. In the last experiment, the effectiveness of the decision model was compared for a different number of extracted features, assuming a constant sequence length of 50 and conducting studies only for the Buffalo dataset. During the analysis, it was shown that the number of extracted features has a significant impact on classification accuracy, achieving the lowest EER coefficient value at 2.36% for an approach in which the feature vector contained features K_1_, K_2_, H1_time_, H2_time_, UD_time_, and DD_time_. However, the authors note that the experiment was conducted for the Buffalo dataset only within the scope of the first task, i.e., the transcription of Steve Jobs’s speech. For other datasets, such as Clarkson II or the second task in the Buffalo dataset, the extraction of the DD_time_ feature may result in a decrease in classification accuracy due to possible long breaks in typing.

The research conducted by Lu et al. in 2020 demonstrated that the use of neural networks in the CNN + RNN architecture as a method for classifying users based on typing patterns allows for promising results, thereby inspiring other researchers for further studies. Building on the approach previously proposed, Kasprowski et al. conducted an analysis of the effectiveness of neural networks in the CNN + RNN architecture, examining the impact of the presence of individual layers and their parameters on classification accuracy in 2022 [23]. The Buffalo dataset was used for the studies; however, in contrast to the base article, the dataset was limited to the second task, i.e., keyboard events recorded during any user activity, which better reflects real user behavior when operating a computer. It is also worth mentioning that the authors decided to apply an overlapping window mechanism, setting the shift coefficient at 40%. The authors’ approach proposed in the paper assumed the construction of a multi-class classifier, assuming that the number of classes is 20, where each class is represented by one user from the Buffalo dataset. The model’s effectiveness was then evaluated depending on the architecture and its parameters. To reduce the temporal complexity of the experiments, data for each user were limited to 1500 keyboard events, subsequently dividing the resulting input set into two sets: training (75% of the input set) and testing (25% of the input set). The study showed that the CNN + RNN architecture allows for higher effectiveness in identifying users based on typing patterns than models using only one type of network (either CNN or RNN). Furthermore, one of the studies conducted by the authors confirmed the conclusion obtained in the base work [22] regarding the optimal sequence length, demonstrating that the correct sequence length for the Buffalo dataset falls within the range of 40 to 60. Additionally, the research extended the architecture of the base model described in the article [22] by adding additional CNN and GRU layers, achieving higher effectiveness compared to the base model. For the proposed model, the anomaly correlation coefficient (ACC) value was 87%. The study also examined the impact of kernel size and the number of filters for the CNN layer on classification accuracy, confirming that the optimal kernel size is 2. Meanwhile, the optimal number of filters, according to the authors, falls within the range of 64 to 256. It is also worth noting that, to reduce the overfitting phenomenon, the authors decided to apply dropout layers between certain layers of the network. According to the authors, the optimal dropout rate is 0.5, indicating a noted decline in the model’s ability to generalize knowledge for lower dropout rate values.

According to the above review of research based on selected key scientific publications in this field, researchers’ interest in user identification based on typing patterns started relatively early. However, only recently have works been conducted that allow for a comparison of results obtained by individual authors. Additionally, the recent development of deep learning techniques has clearly dominated this field and contributed to further improving results achieved in the recent time horizon. These observations inspired the authors of this paper to make their own attempt to define the architecture and build a user verification system, also using deep learning techniques.

## 3. Materials and Methods

### 3.1. Materials

This study aimed to implement a binary classifier for user verification based on keyboard typing behavior. A binary classifier model was developed to understand typing patterns specific to users and estimate the probability of an analyzed key sequence belonging to one of two decision classes. This research focuses on analyzing user interaction with the keyboard interface, extracting features from typing patterns, and employing advanced machine learning algorithms for classification.

To enable the integration of the research findings into the scientific discourse, it was decided to utilize one of the publicly available research datasets representing user typing behaviors, namely the Buffalo dataset [18]. Each of the textual files within the Buffalo dataset represents a chronological sequence of events related to the pressing and releasing of consecutive keys within one session for a single user. An illustrative excerpt from a text file from the Buffalo dataset is presented in Table 1. Each key-related event is represented by two rows, corresponding to the key press and release, respectively, between which another row representing an event for another key may occur, stemming from the pressing of another key before the release of the current one. Following commonly proposed solutions in the literature [13,17,19,22,23], the approach adopted in this study involves grouping keyboard events into digraphs and extracting features from these digraphs. Both numerical and temporal features are considered. For a single digraph, a total of 5 features were extracted:K_1_;K_2_;H1_time_;H2_time_;UD_time_.

It is worth noting that, in the scientific literature addressing these topics, there is no uniformity in the abbreviations used for various features. Alternative abbreviations can be found, e.g., in paper [24]. The features K_1_ and K_2_, respectively, represent the code of the first and the code of the second pressed keys. Key codes in the Buffalo dataset are stored in textual form, implying the necessity of converting the codes from textual to numerical representation. To address the decoding problem, the decision was made to utilize the Label Encoding method. Additionally, to constrain the size of the analyzed data, events pertaining only to a subset of all keys present in the Buffalo dataset were considered, drawing upon the proposed approach in the literature [8]. During the feature extraction process, events for a total of 37 different keys were analyzed, including the following:Keys corresponding to the letters of the English alphabet;Keys corresponding to the digits in the range of 0 and 9;Space key.

The time features H1_time_, H2_time_, and UD_time_ were extracted according to Figure 1.

The attributes DDtime, UUtime, and UDtime have been intentionally disregarded from the analysis. Moreover, to eradicate occurrences related to the continuous depression of the same key, a deliberate decision has been made to eliminate the recorded intermediary values, focusing solely on scrutinizing the press and release times of the specific key. Additionally, to purge the dataset of outliers, drawing upon prescribed values outlined in the literature [25], and acknowledging the potential for negative values to manifest notably in the case of the UD_time_ attribute among proficient keyboard users, both a minimum and maximum acceptable threshold for the UD_time_ attribute have been established, set respectively at −100 ms and 800 ms. Events corresponding to UD_time_ values falling below the stipulated minimum or exceeding the designated maximum threshold have been excluded. An illustrative vector representing the extracted features within a singular digraph is delineated in Table 2.

### 3.2. Methods

The deep learning network architecture proposed by the authors for the decision model implies the necessity of aggregating the extracted feature vectors for consecutively occurring keyboard events into sequences of feature vectors. The selection of an appropriate sequence length is a crucial factor influencing the classification efficacy. Previous studies [22,23] have demonstrated that, for the Buffalo dataset, models based on longer data sequences exhibit higher effectiveness, with the proposed sequence length ranging from 30 to 100. In this study, it was decided to group feature vectors into sequences of length 64. An exemplary sequence of feature vectors of length 64 is presented in Table 3. Additionally, based on the literature [23], the decision was made to employ the overlapping window method, which involves shifting the time window to include a portion of data from the preceding sequence in a specific sequence. The window shift coefficient is expressed as a percentage. In previous studies [23] utilizing the overlapping window method, the shift coefficient was set at 40%. In this study, the coefficient was set at 20%, enabling the inclusion of a greater amount of historical data within the analyzed sequence of keyboard events. An example of applying the overlapping window mechanism within processed data for sequences of length 10 and a shift value of 20% is illustrated in Figure 2.

To construct the decision model, it was decided to utilize neural network systems based on the CNN + RNN architecture, which has been the subject of continuous research in recent years regarding the classification of users based on keystroke pattern analysis for variable text. From the previously published works [19,22,23], it is evident that combining the characteristics of convolutional neural networks with recurrent neural networks leads to an improvement in the obtained results. Figure 3, generated directly using Keras tool, depicts the author’s decision model proposal. The model consists of multiple layers arranged sequentially. However, it is worth emphasizing that selecting an appropriate neural network architecture, as well as determining the suitable parameter values for each layer, requires a thorough analysis of the dataset characteristics and expert knowledge in terms of the construction of classifiers from the deep learning area. Defining the optimal structure of the neural network thus necessitates an extensive review of the research literature in the context of the analyzed problem, defining the dataset characteristics, and thoroughly analyzing the operation of individual neural network layers and their parameters. Nevertheless, some significant aspects concerning the selection of appropriate layers and their parameters within the scope of the analyzed problem have not been addressed in previous works [19,22,23] on the utilization of the CNN + RNN architecture for user classification based on keystroke patterns. Therefore, the architecture of the model proposed in this study was defined based on the conclusions of other researchers presented in previously published scientific papers addressing the problem of user classification based on keystroke patterns, as well as on the authors’ knowledge acquired during the analysis of the subject.

The first of these layers is the input layer, which allows the model to be fed with data. The format of the input data implies the chosen length of sequences N and the length of the vector of extracted features L within a single digraph, which is set to 64 and 5, respectively. Ultimately, the input data of the model form a three-dimensional vector, where the first dimension corresponds to the number of created sequences in the analyzed dataset, the second dimension is equal to the length of a single sequence N, and the third dimension corresponds to the size of the vector of extracted features L.

To enhance the decision model’s ability to generalize knowledge, a 1D convolutional layer (Figure 4) was utilized in the subsequent layer, facilitating the extraction of higher-order features [26]. Among the significant parameters of the convolutional layer influencing the neural network’s effectiveness, particularly considered worthy are the following:The number of filters;The kernel size;The stride value;The activation function.

In this study, the parameter value regarding the number of filters was set at 32 based on the findings of [22], where that number of filters yielded high accuracy in the classification of users based on typing behavior for the Buffalo dataset. The kernel size regulates the width of the feature vector interval analyzed within a given filter, and its value was set at the level proposed in the literature [22,23], which is 2. On the other hand, the stride value is a filter parameter defining the range of movement of that filter within a given sequence. The stride parameter for the model proposed in this study was set to 1 [22]. Selecting an appropriate activation function for neural networks is a complex issue that requires a detailed analysis of the nature of the input data. A commonly used activation function for convolutional neural networks is the Rectified Linear Unit (ReLU) function. However, the ReLU activation function has a significant drawback commonly referred to as the “dying ReLU” phenomenon [27]. In cases where the input value of a neuron is negative, the output value of that neuron for the ReLU activation function is 0, leading sometimes to the complete deactivation of the neuron and the loss of the ability to learn through backpropagation. Considering the characteristics of the input data, specifically the possible negative values for the UD_time_ feature, this study opted to utilize the Leaky Rectified Linear Unit (Leaky ReLU) activation function, which addresses the dying ReLU problem by introducing linearity for negative numbers.

The MaxPooling1D layer, applied after the convolutional layer, allows for a significant reduction in the computational complexity of the model and the extraction of the most significant features obtained during the data processing in the convolutional layer [26].

Following the pooling layer, due to the activation function used in the convolutional layer of the neural network, it was decided to use a data normalization mechanism as the next layer. The Leaky ReLU activation function used takes values from the set (−∞,∞), suggesting the need for data normalization to reduce the size of weights in subsequent layers of the model.

In the subsequent layers of the model, two layers representing a recurrent neural network (RNN) were applied. Popular types of RNNs include Long Short-Term Memory (LSTM) networks and Gated Recurrent Unit (GRU) networks. In this study, it was decided to use GRU networks, referring to previously published works [19,22], which have shown that, for the analyzed dataset, GRU networks allow for higher classification accuracy than LSTM networks.

Between the two GRU layers, a dropout layer was applied to enable the reduction of the overfitting phenomenon, where the dropout rate was set at 50%, based on suggested values from the literature [23].

The final output layer consists of a single neuron, implying the approach adopted in the study for user verification, which involves one-class classification. The sigmoid activation function was used in the final layer, whose range of values belongs to the set of real numbers within the interval [0, 1], modeling the probability function of a given example belonging to the positive class in a binary classification problem [28].

Due to the nature of the decision-making model, which involves binary classification of users based on typing patterns, the binary cross-entropy function was applied as the loss function during the training process of neural networks. Furthermore, to minimize the applied loss function, the widely used Adam optimizer was employed in this study, as commonly utilized in previously published works [23].

## 4. Results

To analyze the efficiency of the proposed approach, for each Buffalo Keystroke Dataset user identifier in the range 001—050, a dedicated model was trained. In total, 50 models were obtained.

The authors of this paper, motivated by the chosen model evaluation methods in [13,19,22], decided to calculate FPR (false positive rate), FNR (false negative rate), and EER (equal error rate) metrics to evaluate the effectiveness of the proposed classification model. Efficiency metrics were calculated for all of the 50 trained models using Stratified Cross-Validation with five iterations, and their final value is an average of all iterations.

Table 4 presents the average value of the each of the calculated metrics for all of the trained models. Additionally, the lowest and the highest values of each metric are contained in Table 4. The obtained results were validated against the results presented in recent related articles. Using different datasets may produce different model evaluation results. Therefore, the authors’ results were compared only to these articles, that also utilize the same Buffalo dataset, keeping the reliability and credibility of the evaluated results.

The average value of the FPR coefficient was accomplished at 1.91%. In so far published articles, the lowest FPR value was obtained at the level of 2.83% [22]. Therefore, a significant improvement compared to the recent state-of-the-art results should be noted. Minimal and maximal values of FPR measure were captured at 0.13% and 6.12%, respectively. In Figure 5, FPR values against every tested user ID model are depicted.

The average value for FNR was recorded at 5.66%, while the lowest value for that measure in recent papers was achieved at 1.89% [22]. The minimal value for the FNR measure was accomplished at 1.91%, and the maximum FNR value was recorded at 12.28%. FNR values, including all tested model results, are presented in Figure 6.

EER coefficient average value, for all trained models, was achieved at 2.65%. In so far published work [22], the lowest value for this measure was established at 2.36%. The minimal and maximal values of the EER measure were accomplished at 0.45% and 6.86%, respectively. Figure 7 depicts EER coefficient values for every trained model against a specific Buffalo user identifier.

The achieved results should be considered outstanding compared to other domain results achieved in examinations that utilize a binary classification approach against the Buffalo dataset (see Table 5). The proposed model, proven by the low value for the FPR coefficient, stands out in terms of keystroke anomaly detection for tested users. Furthermore, knowledge generalization capabilities by the proposed model should be observed, noting that part of every user keystroke sample was left out during the training process of the specific decision model. On the other hand, a limited ability for specific user typing pattern detection should be noted, based on values higher than in other papers for the FNR coefficient. However, the EER coefficient value, close to the recent results in the explored domain, indicates the proposed architecture is promising in terms of further exploration and examination.

## 5. Conclusions

Preceding our own original research with a thorough analysis of the existing literature on the addressed topics not only allowed the starting point for the methods proposed within this article to be defined but, above all, ensured that the best results obtained thus far in this research area were not overlooked.

It is worth noting that the methods developed within the research described in this article are subject to certain limitations, largely stemming from the adopted test dataset. The absence of a widely accessible and universally accepted benchmark dataset remains a significant obstacle in the field of keystroke-based biometric identification. Such a dataset would enable researchers to effectively compare and evaluate the progress of their studies, thus bolstering the credibility and reliability of their findings. This study addressed this challenge by conducting a literature review, wherein authors detailed their most promising results. Consequently, the identification of the most suitable benchmark dataset served as the foundation for the results presented in this paper.

The utilization of keystroke dynamics as a supplementary authentication method alongside traditional approaches enables enhancing its detection threshold for potential intruders. This approach justifies the authors’ focus on improving security, even if it necessitates more frequent reauthorization using conventional methods.

The conducted research has resulted in the development of a method that significantly improves the false positive rate (FPR), with some decrease in the false negative rate (FNR), but without notable impact on the equal error rate (EER). From a practical standpoint, this is a significant achievement, as it is commonly assumed that biometric methods based on keyboard dynamics analysis are not a singular form of securing a given system. This implies that, in cases of uncertainty regarding the user’s identity when using this method, one can resort to reauthorization using a more conventional method, such as a username and password. Achieving a better FPR value thus signifies enhanced security of the protected system, while reducing the FNR represents a cost resulting in a more frequent need for user reidentification.

In summary, this study demonstrates that there is still room for improvement in the quality of identification methods based on keyboard dynamics analysis, and continuous refinement may eventually lead to a more fruitful adoption of these methods in practical applications.

## Figures and Tables

**Figure 1 sensors-24-03763-f001:**
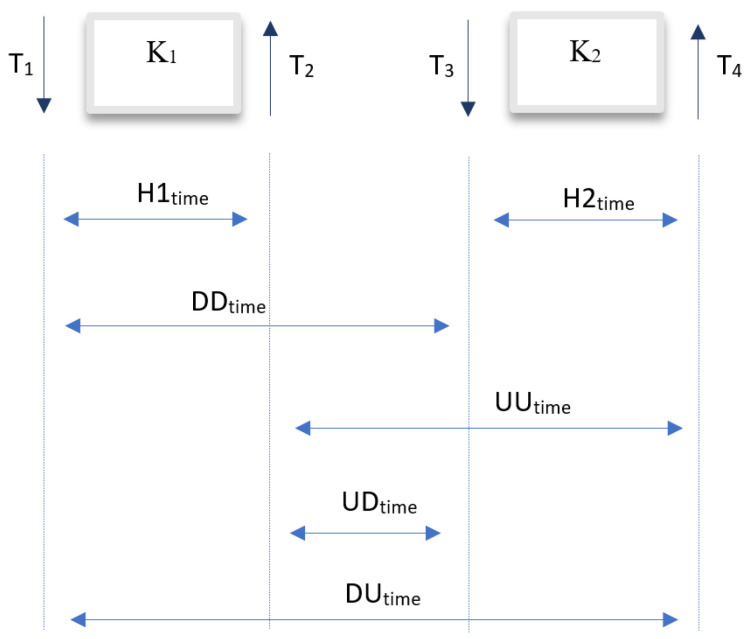
A set of sample features extracted from four consecutive events within two keys.

**Figure 2 sensors-24-03763-f002:**
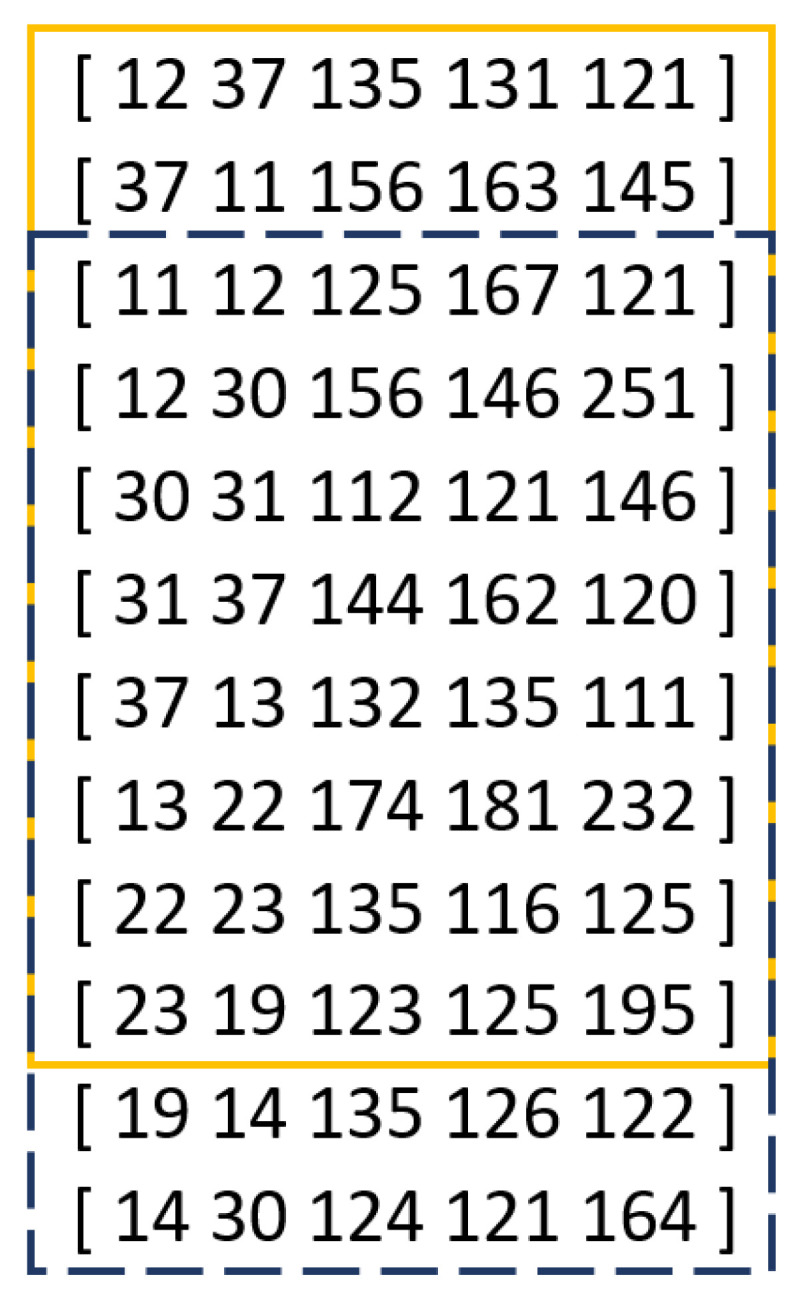
Illustrative example of using the overlapping window mechanism for a sequence of length 10 and an offset value of 20%.

**Figure 3 sensors-24-03763-f003:**
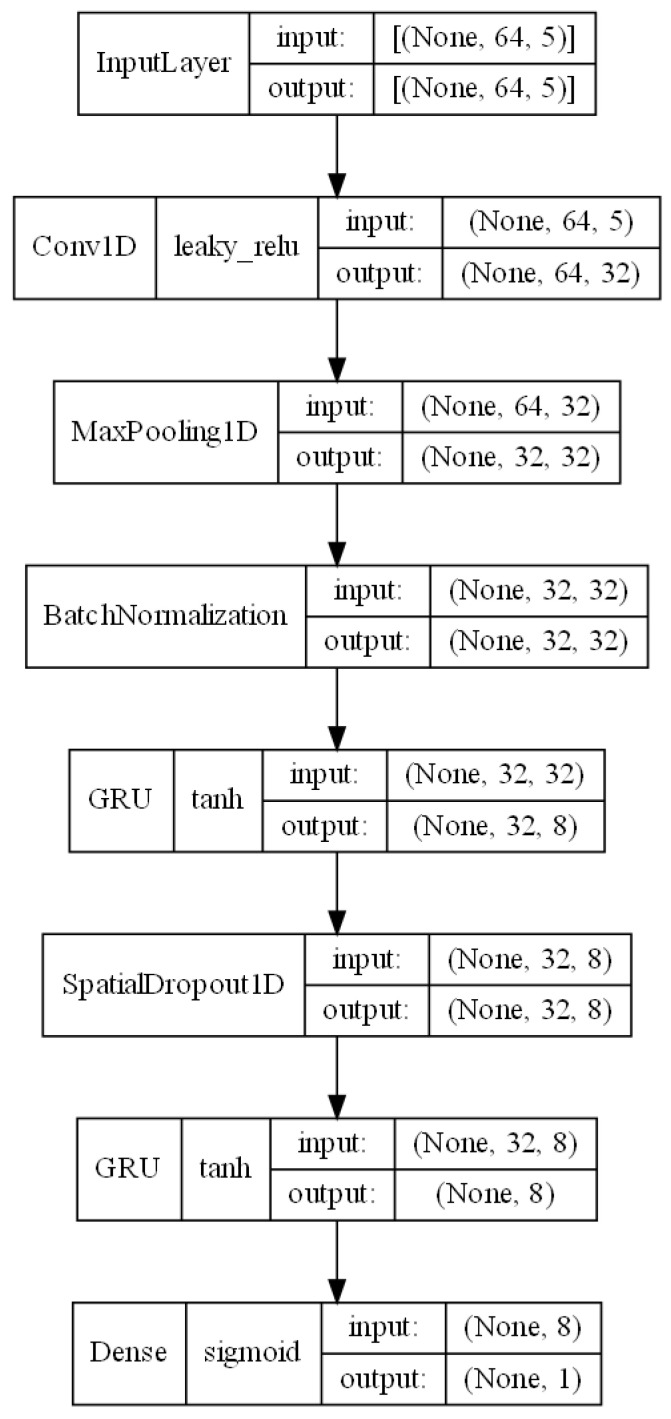
Proposed model architecture.

**Figure 4 sensors-24-03763-f004:**
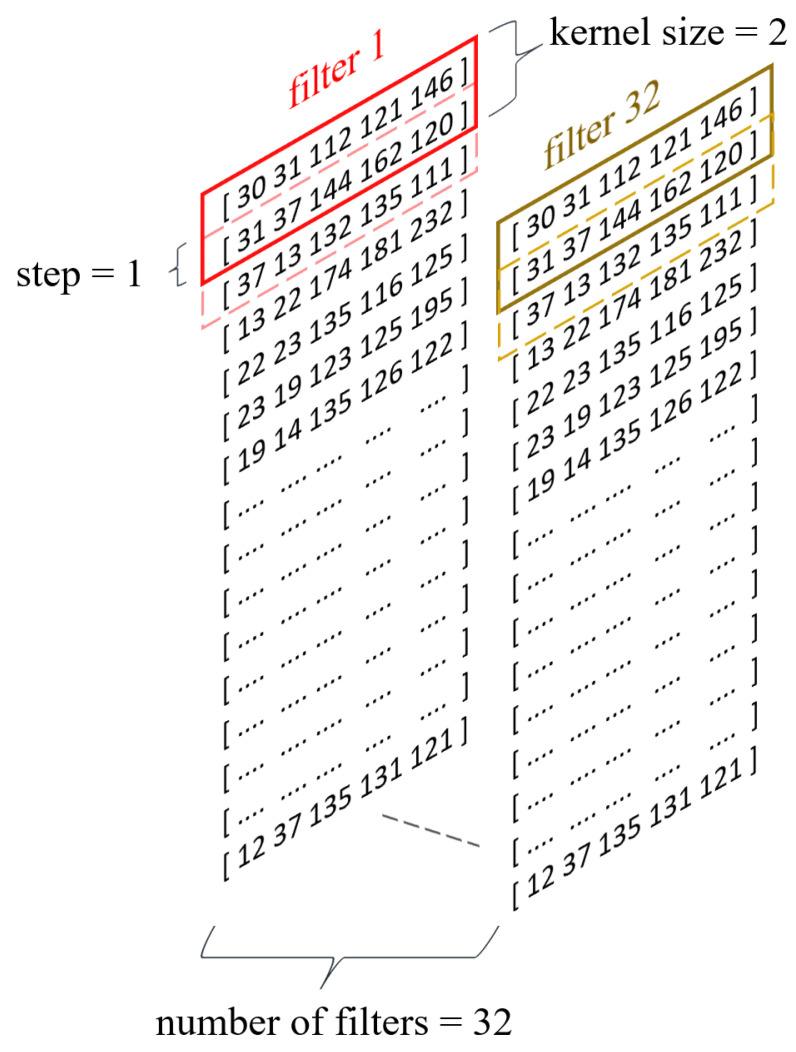
Convolutional layer of proposed model.

**Figure 5 sensors-24-03763-f005:**
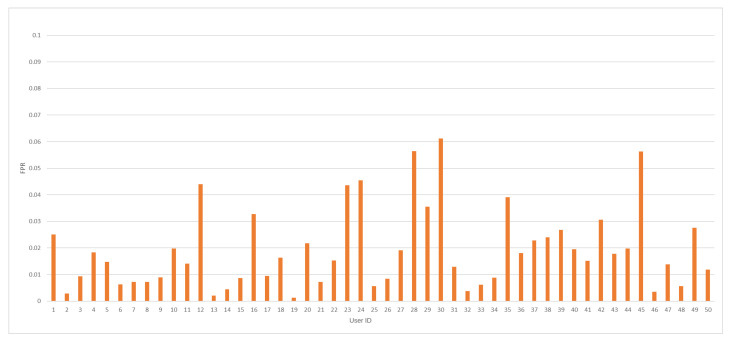
The FPR value distribution by user ID for the analyzed dataset.

**Figure 6 sensors-24-03763-f006:**
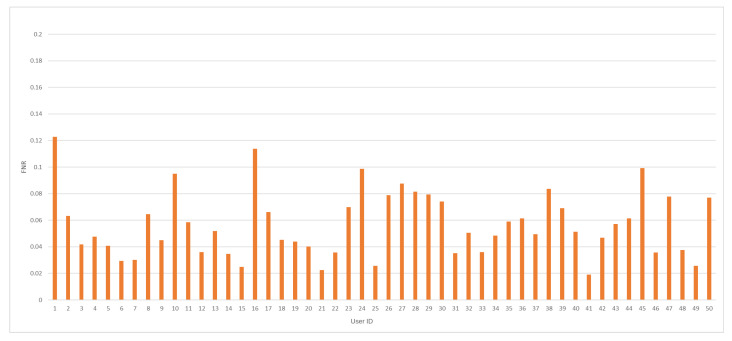
The FNR value distribution by user ID for the analyzed dataset.

**Figure 7 sensors-24-03763-f007:**
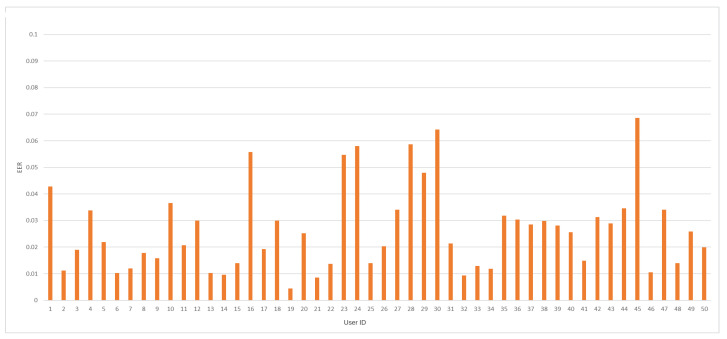
The EER value distribution by user ID for the analyzed dataset.

**Table 1 sensors-24-03763-t001:** Buffalo dataset example file content.

Key Code	Event Type	Timestamp [ms]
A	KeyDown	63578429792961
A	KeyUp	63578429793054
M	KeyDown	63578429793257
M	KeyUp	63578429793382
Space	KeyDown	63578429793429
Space	KeyUp	63578429793554
H	KeyDown	63578429793616
…	…	…
…	…	…

**Table 2 sensors-24-03763-t002:** An example vector representing the extracted features within a digraph.

K_1_	K_2_	H1_time_ [ms]	H2_time_ [ms]	UD_time_ [ms]
12	15	257	327	118

**Table 3 sensors-24-03763-t003:** An example sequence of extracted feature vectors of length N.

Sequence Length	K_1_	K_2_	H1_time_ [ms]	H2_time_ [ms]	UD_time_ [ms]
	12	37	210	181	120
	37	23	181	197	168
	…	…	…	…	…
N = 64	…	…	…	…	…
	…	…	…	…	…
	23	15	199	211	155
	15	9	211	176	147

**Table 4 sensors-24-03763-t004:** Average, lowest, and highest values of the calculated efficiency metrics.

Value	FPR	FNR	EER
average	1.91%	5.66%	2.65%
min	0.13%	1.91%	0.45%
max	6.12%	12.28%	6.86%

**Table 5 sensors-24-03763-t005:** Comparison of obtained results with former achievements.

	FPR	FNR	EER
Authors’ results	1.91%	5.66%	2.65%
Former results [22]	2.83%	1.89%	2.36%

## Data Availability

The original contributions presented in the study are included in the article, further inquiries can be directed to the corresponding author.

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
