# Peer review of "The Improved Biometric Identification of Keystroke Dynamics Based on Deep Learning Approaches"

_sensors, 2024, doi:10.3390/s24123763_

Round 1

Reviewer 1 Report

Comments and Suggestions for Authors

The authors propose a methodology that achieves user authentication by leveraging data derived from the way users type. To achieve their goal, they used a keystroke dynamics dataset, the Buffalo dataset, and deep learning models, CNN and RNN. Their results are good, although they do not manage to surpass the state-of-the-art in this area.

My comments on the work are as follows:

·         “Introduction” is very good. Comprehensive and substantial.

·         I have one objection regarding equation (1). It is not clear if the authors refer to the number of features resulting from keys, digraphs, trigraphs, quadgraphs. Also, the calculations are not correct. For example, for Nk=3 the equation calculates Nc=15, but there are 3 keys (e.g. 'A', 'B', and 'C'), 9 digraphs (e.g. 'AA', 'AB ', 'AC', …, and 'CC'), 27 trigraphs (e.g. 'AAA', 'AAB', …, and 'CCC'), 81 quadgraphs. So, it is a sum of terms Nk, Nk2, Nk3, Nk4, …. I think it's a good attempt at calculating the number of features, but I don't find it correct.

·         “In 2019, Ayotte et al. presented a user classification method based on keystroke dynamics…”. I think is “…authentication method…”. They just used classification algorithms.

·         “Related Work” is much more detailed than it needs to be. Each paper is described extensively, more so than would be done in a Review, and too much detail is given. A lot of information should be removed and it is up to the reader to decide whether they want to expand their knowledge of the specific studies using the references.

·         The names of the features used, namely H1time, H2time and UDtime, are not the usual ones used in keystroke dynamics studies. There is a description to all the different terms used in various studies in “Age and Gender as Cyber ​​Attribution Features in Keystroke Dynamic-Based User Classification Processes”. Perhaps a reference needs to be made.

·         It is not very clear to me why length of feature vectors = 64 was chosen. Wouldn't the stated reason for choosing this value also be satisfied with a value equal to 100, or 80, or even 50? If so, why was 64 chosen. Also, how many key events will the overlapping window move when length=64 and offset value = 20%? I calculate 12.8 key events (feature vectors). Am I doing something wrong in my reasoning? If not, an explanation should probably be given.

·         The methodology is very well described, in simple and explanatory words.

·         I had a little trouble understanding that the 50 dedicated models mentioned are the users used for binary classification. It should probably be written less confusingly.

Reviewer 2 Report

Comments and Suggestions for Authors

The authors examine one approach to authentication that I believe will become increasingly important—behavioral authentication. In this particular case, they investigate Biometric Identification by analyzing Keystroke Dynamics. The great advantage of this approach is that no additional hardware is required for its implementation, and it can be applied to any computer at any time during its use without hindering human activity for identification. I.e. this approach can be used in the background when working with applications requiring special access rights.

The material in the article is presented convincingly and logically. I did not notice any omissions or errors. The obtained results prove the correctness of the chosen approach. The article deserves to be published because of its interesting subject matter and good exposition, although it lacks any breakthrough innovations.

Author Response

Thank you for your positive evaluation of our article. It motivates us to continue our hard scientific work.